# Impact of Vitamin D Deficiency on Mental Health in University Students: A Cross-Sectional Study

**DOI:** 10.3390/healthcare11142097

**Published:** 2023-07-23

**Authors:** Mansour Almuqbil, Moneer E. Almadani, Salem Ahmad Albraiki, Ali Musharraf Alamri, Ahmed Alshehri, Adel Alghamdi, Sultan Alshehri, Syed Mohammed Basheeruddin Asdaq

**Affiliations:** 1Department of Clinical Pharmacy, College of Pharmacy, King Saud University, Riyadh 11451, Saudi Arabia; 2Department of Clinical Medicine, College of Medicine, AlMaarefa University, Riyadh 13713, Saudi Arabia; mmadani@mcst.edu.sa; 3Department of Pharmacy, King Abdulaziz Medical City, Riyadh 14611, Saudi Arabiaalamrial4@ngha.med.sa (A.M.A.); 4Department of Pharmacology, College of Clinical Pharmacy, Imam Abdulrahman Bin Faisal University, King Faisal Road, Dammam 31441, Saudi Arabia; adalshehri@iau.edu.sa; 5Department of Pharmaceutical Chemistry, Faculty of Clinical Pharmacy, Al-Baha University, Al Baha 65528, Saudi Arabia; ai.alghamdi@bu.edu.sa; 6Department of Pharmaceutical Sciences, College of Pharmacy, AlMaarefa University, Riyadh 13713, Saudi Arabia; sshehri.c@mcst.edu.sa; 7Department of Pharmacy Practice, College of Pharmacy, AlMaarefa University, Riyadh 13713, Saudi Arabia

**Keywords:** anxiety, depression, psychological burden, mental health, Riyadh, Saudi Arabia, stress, university students, vitamin D deficiency

## Abstract

Students pursuing a university education are vulnerable to psychological burdens such as depression, anxiety, and stress. The frequency of vitamin D deficiency, on the other hand, is extensively recognized worldwide, and vitamin D regulates various neurological pathways in the brain that control psychological function. Therefore, the goal of this cross-sectional study was to determine the relationship between vitamin D deficiency and psychological burden among university students in Riyadh, Saudi Arabia. During March–May 2021 in Riyadh, a cross-sectional comparative study survey was delivered to university students. The DASS-21 scale was used to determine the severity of the psychological burden. Both univariate and binomial regression analyses were conducted to analyze the level of significance and influence of several factors on the development of psychological burden. The data were analyzed with SPSS-IBM, and a *p* value of <0.05 was considered significant. Of the 480 students recruited for the study, 287 (59.79%) had a vitamin D deficiency. Significantly (*p* = 0.048), a high proportion of the vitamin D-deficient students attained a low or moderate GPA compared to the control cohort. The prevalence of depression, anxiety, and stress among the vitamin D-deficient students was 60.35%, 6.31%, and 75.08%, respectively, which was significantly (*p* < 0.05) different from the control group. The odds of developing depression (OR = 4.96; CI 2.22–6.78; *p* < 0.001), anxiety (OR = 3.87; CI 2.55–6.59; *p* < 0.001), and stress (OR = 4.77; CI 3.21–9.33; *p* < 0.001) were significantly higher in the vitamin D-deficient group. The research shows a strong association between psychological stress and vitamin D deficiency. To promote the mental health and psychological wellbeing of university students, it is critical to create awareness about the adequate consumption of vitamin D. Additionally, university students should be made aware of the likelihood of a loss in academic achievement owing to vitamin D deficiency, as well as the cascade effect of psychological burden.

## 1. Introduction

Vitamin D, a fat-soluble vitamin, plays a key role in maintaining healthy bones [1]. Vitamin D is formed in large amounts by sun exposure. Foods including liver, tuna, and salmon contain a modest quantity of vitamin D [2]. Researchers have shown a correlation between vitamin D deficiency and mortality, heart failure, and myocardial infarction [3]. A lack of vitamin D has been linked to a number of health issues, including cancer, osteoporosis, autoimmune illnesses, mental disorders, respiratory problems, and osteomalacia [4,5,6]. The extensive expression of vitamin D receptors in many organ systems has been implicated in the widespread systemic effects of vitamin D [4,7].

Lower serum vitamin D levels have been linked in several studies to psychological distress [8]. Multiple studies have indicated that vitamin D has beneficial effects on neurocognitive function [9,10]. The prevalence of vitamin D deficiency has been estimated to be over one billion persons worldwide [11]. However, the incidence varies greatly from one country to another. It is also of remarkable importance throughout the Middle East and Asia, despite the region’s year-round sunshine [12,13].

Numerous studies have assessed the significant prevalence of vitamin D deficiency among Saudi individuals. Elshafie et al. [14] carried out a study on vitamin D deficiency in Saudi Arabia among 50 Saudi married couples and found that 70% of women had the condition compared to 40% of males. Another study [15] revealed that 71% of female medical students were vitamin D deficient. However, research from the Qassim Region of Saudi [16] found that just 28% of the population was vitamin D deficient.

Psychological distress and mood swings are just two of the many elements that make up mental distress. As defined by the American Psychiatric Association in its 1994 *Diagnostic and Statistical Manual of Mental Disorders*, depression is characterized by a persistent and pervasive state of sadness or a lack of interest or pleasure in virtually all activities for at least two weeks. The pathophysiology of mental disorders has been poorly understood because of the wide variety of clinical presentations and underlying causes [17]. According to Vieth et al. [18], mental disorders are complicated illnesses with a variety of subtypes and causes, including a potential vitamin D role. Numerous areas of the brain have been shown to contain vitamin D receptors. These receptors have developed in parts of the brain that are connected to the onset of psychological illness. Because of this, vitamin D has been connected to mental health issues [19].

The pathogenesis of psychological distress is not clearly known, and it is likely that several diverse mechanisms are at play despite the biological, psychological, and environmental explanations that have been put forward [20,21,22,23]. Wilkins et al. [24] and May et al. [25] found a significant association between low vitamin D levels and psychological distress; however, Chan et al. [26] and Pan et al. [27] found no association.

Przybelski and Binkley’s [28] retrospective chart review study revealed the effects of vitamin D deficiency on memory function and cognitive decline. Despite numerous research studies claiming a link between vitamin D shortage and cognitive impairment, there is currently a lack of data demonstrating that a lack of vitamin D has any impact on academic performance [29]. However, a recent study [30] refutes the association between low vitamin D levels and academic performance.

Numerous research studies have linked vitamin D deficiency to psychological disturbance, as was previously indicated. However, there is not a study in the literature that looks at how these two variables relate to university students in Riyadh, Saudi Arabia. With that in mind, the current study sought to employ a standardized, validated DASS scoring system to better understand how vitamin D deficiency affects the mental health of students at AlMaarefa University, Riyadh, Saudi Arabia.

## 2. Materials and Methods

### 2.1. Study Design

This research was conducted using a cross-sectional comparative study design with a validated questionnaire carried out among students from AlMaarefa University, Riyadh, from March to May 2021. The time corresponded to the period when students returned to the on-campus education system from the COVID-19-mediated virtual system of teaching. It involved a comparison of the psychological burden with vitamin D deficiency status based on their affiliated college and socio-demographic features.

### 2.2. Sampling

The participants in our study were AlMaarefa University students affiliated with any of the three colleges (College of Medicine, College of Pharmacy, and College of Applied Science), aged above 18 years, who had provided consent to participate. They were recruited using simple random sampling; the randomly selected subjects were approached, and the interviewer administered questionnaires to those who consented to participate in the study. Participation in this study survey was entirely voluntary, and complete confidentiality and anonymity were maintained, with no identifying information being recorded in the survey results. A consent form was added at the beginning of the questionnaire explaining the purpose of the study, the objectives, a description of the research project, and a request for their participation. To confirm the patients’ vitamin D status, the subjects’ vitamin D levels were measured using an enzyme-linked immunosorbent assay (ELISA) machine from BioTek and a vitamin D 25-OH ELISA assay kit from Calbiotech Incorporation. In this study, participants were deemed to be vitamin D-deficient if their 25(OH)D levels were under 20 ng/mL [16]. The Institutional Review Board of AlMaarefa University, Riyadh, approved the study protocol (202/12/RC, 19 September 2020).

### 2.3. Study Questionnaire

The data for the study were gathered using a validated structured questionnaire. The questionnaire was developed after extensive literature research and consulting with specialists. Following the validation process, the questionnaire was used in a pilot study involving thirty independent study samples. The questionnaire was refined and fine-tuned based on comments from pilot research participants. There were three sections to the questionnaire. The demographic information in the first section included the age, gender, colleges/departments, study level, and nationality of the participants. The GPA of the enrolled students was obtained from the university register.

The second section of the questionnaire included items that evaluated students’ overall characteristics. Recent weight changes, self-perception of body shape, level and duration of physical activity, sun exposure, presence or absence of vitamin D deficiency, use of sun protection, use of artificial vitamin D sources such as tanning beds, vitamin D intake from food, risk factors for depression, and use of antidepressants were all examined.

To determine the prevalence of psychological load among research participants, a depression, anxiety, and stress scale that had undergone extensive international testing and approval was used. The severity of several symptoms that are typical of depression, anxiety, and stress was evaluated using a 21-item self-report questionnaire called the DASS 21. Items of the DASS-21 mentioned at positions 3, 5, 10, 13, 16, 17, and 21 were meant to assess the depression status, while item numbers 2, 4, 7, 9, 15, 19, and 20 were able to detect anxiety in the respondents. Furthermore, 7 other items (1, 6, 8, 11, 12, 14, and 18) were used to determine the level of stress in the surveyors. The person needed to mention whether a symptom was present throughout the previous week when completing the DASS. Each item was given a score between 0 (which did not apply to the participant over the last week) and 3 (meaning it did apply to the participant frequently). The answers were based on a Likert scale with 0 referring to “never”, 1 referring to “sometimes”, 2 referring to “often”, and 3 referring to “almost always”. After reducing the traditional scale of 42 items, this section now featured 21 items. To obtain the final score, the ratings of depression, anxiety, and stress scores were added together and multiplied by two. Henry and Crawford [31] divided the scales into mild, moderate, and severe categories to rate each state’s severity. For mild, moderate, and severe depression, the cutoff values were 10, 14, and 21, respectively. For mild, moderate, and severe anxiety, the cutoffs were 8, 10, and 15, respectively. Finally, the cutoff values for mild, moderate, and severe stress were 15, 19, and 26, respectively [32]. Similarly, scores of 10, 8, and 15 in their respective items were regarded as signs of depression, anxiety, and stress, respectively [33].

To ease the understanding of the questionnaire, the validated English questionnaire was translated into Arabic using a forward–backward method with the help of subject experts with good command over both languages. The validated Arabic version of the DASS-21 items was included in the bilingual form of the questionnaire [34]. The Arabic version was translated back to English to ensure the accuracy and uniformity of the questionnaire contents in both languages.

### 2.4. Data Collection

The data collection team consisted of the Pharm.D program students from the College of Pharmacy, AlMaarefa University, Riyadh, supported by senior faculty members. This team was trained in introducing the study subject to the randomized participants, presenting them with the bilingual survey forms, and collecting their responses. The participants were briefed about the purpose, procedures, and potential risks in the Arabic language, and they consented by ticking the “agree to participate” column on the first page of the questionnaire.

### 2.5. Statistical Analysis

Statistical Software for Social Science, version 23 (IBM SPSS Inc., Chicago, IL, USA), was used to enter and analyze the acquired data. Frequencies and percentages were used appropriately to present the data. Academic achievement and the students’ answers to questions about their vitamin D status were compared. To further investigate the role of vitamin D insufficiency in the development of psychological distress, the DASS score was compared with vitamin D deficiency status. To determine the statistical significance of variations in proportions of categorical data, Pearson’s Chi-square and Fisher’s exact test (two-tailed, if appropriate) were performed. To examine the association between sociodemographic factors, vitamin D deficiency status, and different forms of psychological burdens, binary logistic regression analysis was carried out to obtain the odds ratios (ORs) and 95% confidence intervals (CIs). *p* values lower than 0.05 were regarded as significant.

## 3. Results

### 3.1. Demographic Characteristics

The study included 480 university students, and a significantly (*p* = 0.032) high proportion (Table 1) was between the ages of 20 and 22 years. Compared to their male counterparts, a slightly higher proportion of female students (46% vs. 54%) was included in the study. The number of students enrolled from the university’s three colleges was similar, while the number of students from the College of Medicine was slightly less than that from the other two colleges. The proportion of students in the middle level of their studies (5th to 7th level) was 39.37%, which was not statistically different than the other levels (1–4 level and 8 level). The institution offers two semesters in a single academic year (except the summer semester, typically chosen by repeaters), with each level of study offered in a single semester. Most of the respondents who participated in our research were Saudi nationals (81.45%), and a significantly high number of the participants had grades ranging from 1.5 to 3.4 on a four-point scale.

### 3.2. General Characteristics of the Participants

A large percent (60%) of participants reported having their body weight significantly fluctuate over the preceding three months (weight loss: 32.08%; weight gain: 28.54%), whereas 39.37% of participants believed they maintained the same body shape. The percentage of students who thought of themselves as slim, normal, and obese was 18.13%, 48.13%, and 33.75%, respectively. Regarding physical activity, fairly similar percentages were found for people who engaged in physical activity (48.13%) and those who did not (51.88%). Of the 48.13% of people who exercised physically, 23.33%, 25.83%, and 8.96% of people exercised one to two days per week, three to four days per week, and more than four days per week, respectively. Participants’ descriptions of the physical activity patterns ranged from vigorous for at least 20 min (27.5%) to moderate for at least 30 min (14.38%) to walking for at least 30 min (6.25%). Most individuals (48.13%) spent less than one hour outside each day, whereas 25.42% spent between one and two hours outside each day. Furthermore, 16.9% of people spent 2–4 h outside daily compared to 7% who spent between 4 and 6 h outside daily. A significantly (*p* = 0.021) high percentage (59.79%) of the individuals reported having vitamin D deficiency, whereas the remaining 40% either did not have or did not know if they had any vitamin D deficit. A very small percentage of participants (37.92%) were using vitamin D supplements or multivitamins containing vitamin D, compared to a large proportion (62.08%) who denied taking any vitamin D supplements. Most students who participated in the study said they had not used sun protection cream in the previous 12 months (63.33%) or tanning beds (66.25%). Many students (391, 81.46%) who participated in our survey denied using any antidepressant medication, while just 18.54% were taking antidepressants (Table 2).

### 3.3. Vitamin D Status and Academic Performance

Table 3 shows that there was a significant (*p* = 0.048) association between vitamin D deficiency and the academic achievement of the student. When compared to students who did not have any known vitamin D deficiency or who did not use vitamin D supplements, students with known vitamin D deficiencies had significantly (*p* = 0.017) poor GPAs. GPA and the study participants’ use of tanning beds or sunblock were not shown to be significantly correlated with one another.

### 3.4. Symptoms of Vitamin D Deficiency

The potential signs and symptoms that a vitamin D deficiency can produce are shown in Figure 1. Vitamin D deficiency is just one of many potential explanations for these symptoms; yet, in some cases, the development of these symptoms may lead to the unexpected diagnosis of vitamin D deficiency. Muscle weakness and chronic fatigue are the most observed symptoms. These symptoms necessitate the investigation of other potential causes and evaluating vitamin D levels to rule out deficiency.

### 3.5. Dietary Status of Participants

Figure 2 shows the percentage preference for diets rich in vitamin D among our study samples with vitamin D deficiency. Cheese was the dominant selection for a diet rich in dairy products, which was followed by fatty fish and egg yolk. People with a regular intake of vitamin D-rich food may not develop vitamin D deficiency and its complications.

### 3.6. Risk Factors for Depression

Figure 3 depicts the possible risk factors for the induction of depression. Excessive academic demand (21%) was seen as the most common risk factor, followed by a family background of psychological burden (18%). Other factors selected by our study participants were thyroid diseases (14%), heart diseases (11%), the recent death of someone in the family (11%), and social isolation (9%). Some participants selected factors such as job loss, cancer, family disputes, drug abuse, and chronic injury.

### 3.7. Drug Use Profile That May Cause Vitamin D Deficiency

Figure 4 depicts the profile of responders exposed to medicines or agents that could cause vitamin D deficiency. An extremely low proportion of respondents said they routinely use one of the listed agents. Isotretinoin had the most significant percentage (8.9%), and given that most individuals were within an age range where acne is typically more prevalent, it may have been recommended for treating acne. Since more than 33% of research participants were obese, 3.8% of surveyors acknowledged using statins. Among the respondents, only 3.1% reported using opioids, beta-blockers, or benzodiazepines. Varenicline (a smoking cessation medication) and alcohol were utilized by only 2.1% of the students. Only 1.7% of the subjects reported using calcium channel blockers or nuvaring (to manage pregnancy). Acyclovir, interferon, and anticonvulsants were only used by 0.3% of those surveyed.

### 3.8. Analysis of Psychological Burden Using DASS-21

Table 4 compares the profile of psychological burden of participants with vitamin D deficiency with the control group. The DASS-21 was used as a scoring system for determining the prevalence of psychological burden. A significantly (*p* = 0.042) high percentage of the vitamin D-deficient students was found with depression (60.35%) compared to the control group (47.66%). Overall, 55% of those surveyed in this study were diagnosed with having depression symptoms. Further, Table 4 shows that the prevalence of anxiety in our sample of students was 60.83%. There was a significant (*p* = 0.031) difference between the vitamin D-deficient and the control groups regarding the prevalence rate of anxiety. Additionally, DASS-21 helped measure the status of stress among the participants. The percentage of vitamin D-deficient patients (75.08%) under mental stress was significantly more than the control group (58.03%). The overall stress prevalence was 67.91% among the students included in the study.

### 3.9. Level of Severity of Psychological Burden among Participants

As shown in Table 5, around 70% of the vitamin D-deficient participants in our study were suffering from either moderate or severe forms of depression (*p* = 0.032) compared to the control, which was only 53%. Further, a severe form of anxiety was significantly (*p* = 0.012) more common (13.75%) in vitamin D-deficient students compared to the control cohort (5.82%). Significantly (*p* = 0.011) a higher proportion of the vitamin D-deficient students was found with either moderate (59.2%) or severe (15.88%) forms of stress compared to the control group of the participants.

### 3.10. Regression Analysis for Depression

As per the details given in Table 6, the risk estimate to develop depression was highest due to vitamin D deficiency (OR = 4.96, CI 2.22–6.78, *p* = 0.001), being female (OR = 3.45), experiencing muscle pain (OR = 3.21), frequent tiredness (OR = 2.98), excessive academic demand (OR = 2.87), family history of psychological burden (OR = 2.76), obesity (OR = 2.67), and use of anti-acne medicine (isotretinoin) (OR = 2.34) were other significant factors that influenced the chances of developing depression among the study participants. Using statin showed a decreased risk (OR = 0.342) for developing depression, whereas 20–22 years and early study level (between 1 and 4) had increased odds of depression of 1.67 and 1.32 times, respectively.

### 3.11. Regression Analysis for Anxiety

As depicted in Table 7, a family history of psychological burden showed the highest risk (OR = 4.55, CI 2.25–8.65, *p* = 0.001) for developing anxiety among the study participants. Further, patients who experienced frequent tiredness, and those who used anti-acne medicine (isotretinoin), experienced increased odds of anxiety development of 4.21 and 4.01 times, respectively. Vitamin D deficiency was found to be the fourth risk factor (OR = 3.87, CI 2.55–6.59, *p* = 0.001) for anxiety in our study samples. Excessive academic demand (OR = 3.56), the experience of muscle pain (OR = 3.22), female gender (OR = 3.21), obesity (OR = 3.11), age group 20–22 years (OR = 2.13), and early study level (1–4 level) (OR = 1.32) were other significant factors that increased the risk for the development of anxiety among the surveyed university students. Interestingly, the use of statin (OR = 0.54), benzodiazepines (OR = 0.44), and beta blockers (OR = 0.32) showed significantly less risk for anxiety compared to those who were not using those medicines.

### 3.12. Regression Analysis for Stress

Family history of psychological burden had the highest risk (OR = 5.67, CI 2.071–9.32, *p* = 0.001) of causing stress among the study participants (Table 8). Anti-acne medicine (isotretinoin) was associated with the second highest risk (OR = 4.87, CI 2.98–8.64, *p* = 0.001) of developing stress among the university students who participated in this study. The odds of having stress were greater in vitamin D-deficient individuals by 4.77 times, whereas obese individuals had an increased risk of stress by 4.21 times. Experiencing frequent tiredness (OR = 3.44), excessive academic demands (OR = 3.21), experiencing muscle pain (OR = 2.45), female gender (OR = 2.33), study level (1–4) (OR = 2.31), and age (20–22 years) (OR = 1.32) were significantly associated with an increased chance of developing stress among our study samples. Those participants who were using statins (OR = 0.89), benzodiazepines (OR = 0.87), and beta blockers (OR = 0.56) had a lower chance of stress induction compared to those who were not using these medicines.

## 4. Discussion

This study was conducted to determine the association between deficiency in vitamin D and psychological burden in university students of Riyadh, Saudi Arabia. Further research was conducted to ascertain the association between academic achievement and the status of vitamin D deficiency. We also investigated additional variables that affect the emergence of psychological loads and compared their impact with that of vitamin D deficiency. The results of the inferential analysis point to a considerable impact of vitamin D shortage on the occurrence of depression, anxiety, and stress, as well as a clear correlation between vitamin D deficiency and a decline in academic performance.

The prevalence of vitamin D deficiency in the current study (59.794%) is lower than that reported by the authors [15], where (70.7%) of their sample reported having a vitamin D level below 20 ng/mL, even though the sample also included healthcare students, as in our study. In addition, most participants in our study (62.08%) denied taking vitamin D supplements. This difference in vitamin D deficiency prevalence could be related to the participants’ healthy lifestyle choices; a sizable portion of our study samples (39.37%) had a normal body shape and exercised regularly (48.13%), at least 1–2 days per week.

We found in our study that there was an association between vitamin D deficiency and academic achievement, where we found that there was significantly low academic achievement among vitamin D-deficient students. These results contrast with a study that found no connection between low vitamin D levels and academic performance [29]; however, another study found that over 90% of students who did not consume enough vitamin D were more likely to perform slightly less well academically than those who consumed adequate amounts of vitamin D and other dietary sources [30]. A recent study [35] conducted in Saudi Arabia demonstrated that there was a drop in the academic performance of health science university students in the presence of vitamin D deficiency. Therefore, as our study participants are mostly health science students, it is possible that subject overload in health science programs is further increasing the workload on the already-vulnerable vitamin D-deficient students, thereby resulting in a significant decrease in the academic attainment of our study participants.

Although bone fracture is one of the main symptoms of vitamin D deficiency [36], it was the least noticeable symptom in our study populations. The participants’ ages ranged from 20 to 26 years, and bone fractures are less obvious in young people than in the elderly, which may account for the notable difference in results between these symptoms. According to Feskanish et al. [37], hormonal changes that may decrease bone density may cause bone fractures to occur more frequently in elderly women (postmenopausal women) with vitamin or calcium deficiencies than in younger women.

The majority of the participants’ diets consisted of cheese and other dairy items, followed by diets rich in fatty fish and egg yolk. People who consume these diets on a regular basis may not develop vitamin D deficiency, and this may be one of the possible reasons for having a relatively lower percentage of vitamin D-deficient individuals, as demonstrated previously by a systemic review study conducted by Bolland [38].

In addition to being a food, vitamin D is also a hormone that has receptors in almost all the body’s cells and tissues. It has been demonstrated that vitamin D supports healthy physical, mental, and immunological system function and has a wide range of effects on systemic health. Correlational scientific evidence continually demonstrates an adverse association between low vitamin D levels and mental health issues, including depression and anxiety, across all age groups [39]. The findings in our study show that there is a significant influence of vitamin D deficiency on developing depression, anxiety, and stress among young university students. The risk of depression increased by five times in vitamin D-deficient students compared to the control cohort. These findings are in agreement with a meta-analysis of 31,424 participants, which showed a significantly high risk of depression in participants who had lower than normal vitamin D levels [21]. Another study with 7970 participants indicated that the odds ratio for depressive episodes is considerably higher in those with serum vitamin D levels below 20 ng/mL compared to those with levels above 30 ng/mL [40]. In addition to depression, the risk of anxiety increased by 3.87 times, and the odds of stress were enhanced by 4.77 times in vitamin D-deficient patients compared to the normal participants.

The hypothalamus pituitary adrenal (HPA) axis is a dynamic feedback loop between the central nervous system and the endocrine system that is activated in response to stress. Anxiety and mood disorders, as well as other mental health problems, have been linked to the malfunctioning HPA axis. When exposed to UVB light, the skin has a systemic effect on the HPA axis, ensuring appropriate vitamin D levels [41,42]. Studies conducted in vitro and on animals have demonstrated that skin exposure to UVB light causes the expression of all HPA axis components, including corticotropin-releasing hormone, proopiomelanocortin, adrenocorticotropic hormone, beta-endorphin with associated receptors, the glucocorticoidogenic pathway, and the glucocorticoid receptor [41]. Dysfunction of this activation may result in a variety of mental stresses and the development of anxiety, so vitamin D deficiency is eventually a cause of psychological distress. It is also important to note that calcitriol, the active form of vitamin D, drives gene transcription in the brain that functions to both induce serotonin synthesis and block reuptake, likely increasing serotonin levels in the central nervous system [43,44]. Therefore, it is thought that maximizing vitamin D may aid in preventing and reducing the severity of brain dysfunction.

One of the major contributing factors to the development of psychological burden in our study samples was the female gender. There are several reports that validate the relationship between the female population and the occurrence of psychological burdens [45]. According to a study conducted in Egypt, girls are more likely than boys to experience significant levels of depression and anxiety [46]. A study conducted in Saudi Arabia reported that the prevalence rate of psychiatric disorders is in the range of 30 to 46% [47]. Further, we found that vitamin D deficiency was strongly associated with the female gender in our study samples, which is similar to earlier reports [8,48], and we attribute this to prevalent local cultural norms that limit skin exposure to sunlight [49]. Therefore, with vitamin D deficiency, women are more likely to develop psychological distress. This was confirmed by Bassil et al. [8] in a systematic review of vitamin D prevalence and predictors in the Middle East and North Africa region. They concluded that despite the region’s high sunshine levels, hypovitaminosis D is extremely common, with a prevalence of between 30% and 90%, and adult risk factors include older age, female sex, multiparty, the season of the year, style of clothing, socioeconomic status, and residence (urban rather than rural). Therefore, there is a need for necessary steps to improve the vitamin D level, which will considerably improve the mental status of the population in general and females in particular.

The use of statins, benzodiazepines, and beta blockers was inversely correlated with the occurrence of psychological burden in the study population. Our findings are congruent with a meta-analysis that reported statistically significant improvements in mood scores among the 2105 participants [50]. The benefit of statin could be attributed to its anti-inflammatory, antioxidant, and cardioprotective properties [51]. Benzodiazepines are psychoactive agents known for their anxiolytic effects [52], and hence we found in our study samples a decreased correlation between stress and anxiety with the use of this drug. Nevertheless, the use of this drug was found with only 3.1% of the population, which indicates the cautious approach of our study participants to deal with this drug due to its drug dependence and side effects. Further, the use of beta-blockers was associated with decreased vulnerability to stress and anxiety among the students who participated in this study. Our findings are similar to those of an earlier study [53], where they observed links between beta-blockers and lower psychological distress.

Despite some intriguing findings, our study has several limitations. First, because it was cross-sectional, we were unable to determine causality. Additionally, this could have exposed the study to sources of bias due to disparities in the participants’ cultural backgrounds, ages, and socioeconomic levels, as well as bias originating from the way study subjects were recruited. However, as a sizable portion of the students attending AlMaarefa University are from other parts of the country, the participants in the current study were not restricted to the capital city of Riyadh. This renders the selection of cases impartial and perhaps representative of Saudi society. Second, we conducted our research from March to May of 2021, which is the start of the summer. The research area’s temperature is generally always between 25 and 35 degrees Celsius during this time, and the sun is not as intense as it is from June to August; nevertheless, considering the availability of sunshine nearly all year, seasonality may not be a relevant effect in our instance.

## 5. Conclusions

Even though there is ample sunlight in Saudi Arabia, vitamin D deficiency is widespread among the population. The need for increased vitamin D awareness, particularly among university students, and the integration of vitamin D testing in primary healthcare facilities, vitamin supplements, and foods fortified with vitamin D are necessary, particularly for people who are dealing with psychological burdens such as depression, anxiety, and stress. Controlling vitamin D levels not only helps to control the wide range of psychological burdens but also helps to improve the academic performance of the students. It is especially critical to provide additional coverage for the female population to combat the high frequency of both psychological discomfort and vitamin D deficiency.

## Figures and Tables

**Figure 1 healthcare-11-02097-f001:**
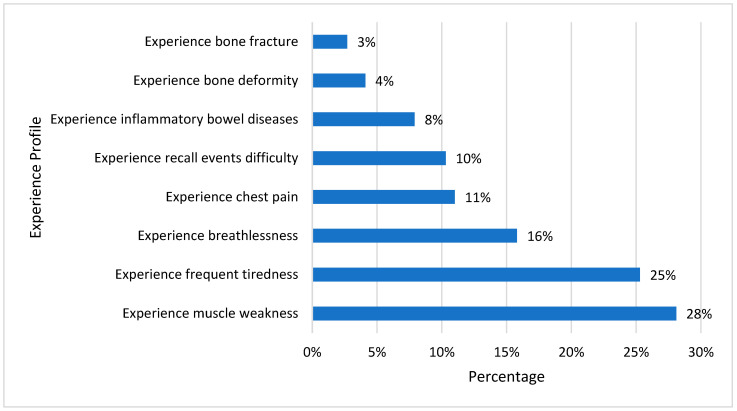
Experience profile of vitamin D-deficient participants in the last 12 months.

**Figure 2 healthcare-11-02097-f002:**
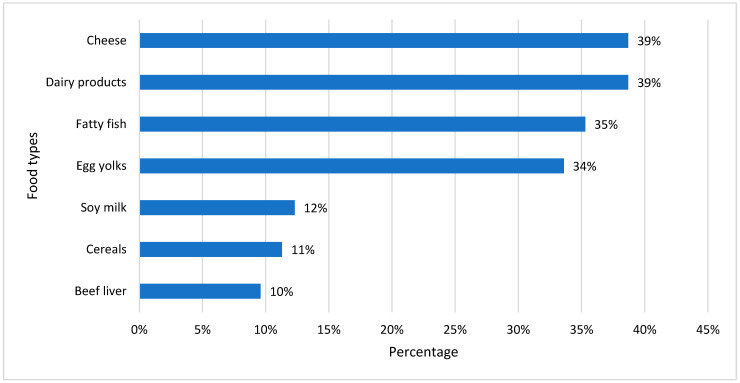
Common food habits of vitamin D-deficient participants.

**Figure 3 healthcare-11-02097-f003:**
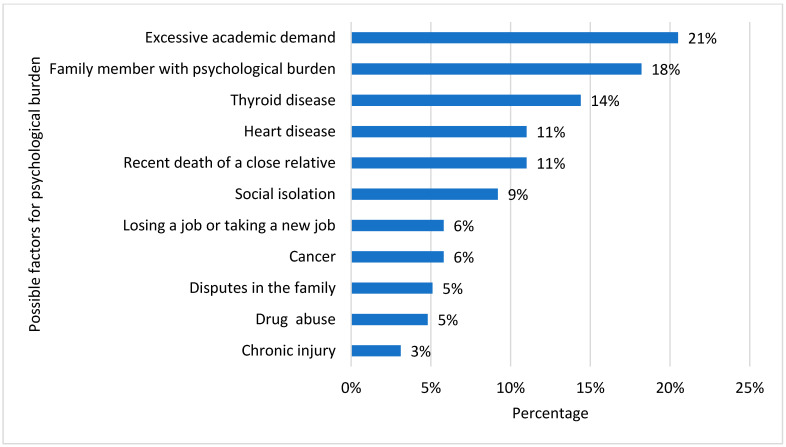
Common depression determinants in vitamin D-deficient participants.

**Figure 4 healthcare-11-02097-f004:**
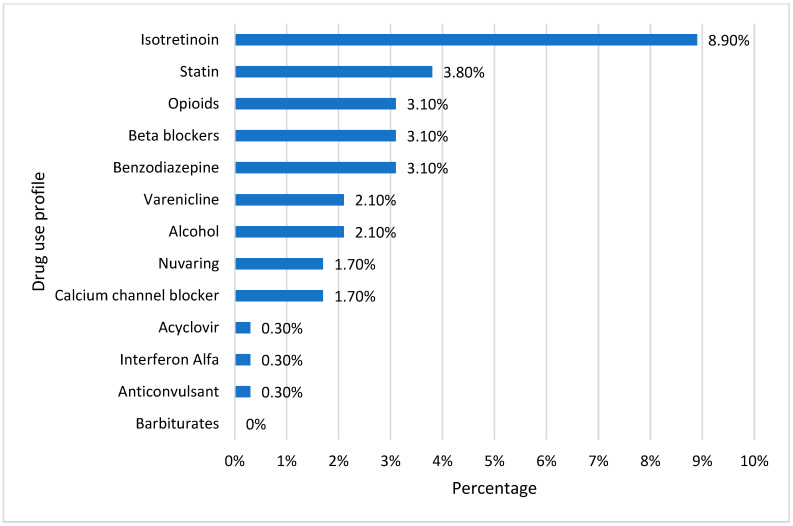
Drug use profile for causing vitamin D deficiency.

**Table 1 healthcare-11-02097-t001:** Demographic characteristics of the participants.

Characteristics	Variables	Frequency (n = 480)	Percentage	*p* Value
Age	20–22 years	245	51.04%	0.032
	23–25 years	134	27.91%	
	>25 years	101	21.04%	
Gender	Male	221	46.04%	0.654
	Female	259	53.95%	
College	College of Medicine	135	28.12%	0.078
	College of Pharmacy	182	37.91%	
	College of Applied Science	163	33.95%	
Study level	1–4	122	25.41%	0.754
	5–7	189	39.37%	
	≥8	169	35.20%	
GPA (in a scale of 4)	0–1.5	137	28.54%	0.048
	1.51–3	215	44.79%	
	3.1–4	128	26.66%	
Nationality	Saudi	391	81.45%	0.001
	Non-Saudi	89	18.54%	

**Table 2 healthcare-11-02097-t002:** General characteristics of the subjects.

Characteristics	Variables	Frequency	Percentage	*p* Value
Did you gain or lose body weight during the last three months?	Weight loss	154	32.08%	0.432
	Maintaining	189	39.37%	
	Weight gain	137	28.54%	
Your perceived body shape	Lean (slim)	87	18.13%	0.094
	Normal	231	48.13%	
	Obese	162	33.75%	
Do you practice physical exercise?	Yes	231	48.13%	0.765
	No	249	51.88%	
How many times do you do physical exercise per week?	1–2 days per week	112	23.33%	0.065
	3–4 days per week	76	15.83%	
	More than 4 days per week	43	8.96%	
	Not applicable	249	51.88%	
What is your duration of physical exercise per week?	Intense physical activity for at least 20 min	132	27.50%	0.076
	Moderate physical activity for at least 30 min	69	14.38%	
	Walking at least 30 min	30	6.25%	
	Not applicable	249	51.88%	
Including exercise, how many hours do you spend outside during daylight hours?	Less than 01 h/day	231	48.13%	0.034
	1–2 h/day	122	25.42%	
	2–4 h/day	81	16.88%	
	4–6 h/day	38	7.92%	
	More than 6 h	8	1.67%	
Do you have vitamin D deficiency?	Yes	287	59.79%	0.021
	No	98	20.42%	
	I do not know	95	19.79%	
Do you take vitamin D supplement or a multivitamin that includes vitamin D?	Yes	182	37.92%	0.038
	No	298	62.08%	
Have you used sunscreen/sun protective cream in the last 12 months?	Yes	176	36.67%	0.041
	No	304	63.33%	
Did you ever try tanning beds/tanning booths?	Yes	162	33.75%	0.042
	No	318	66.25%	
Do you currently take an antidepressant medication?	Yes	89	18.54%	0.001
	No	391	81.46%	

**Table 3 healthcare-11-02097-t003:** Association between vitamin D status and academic performance based on student GPA (grade point average).

Questions	Variables	GPA, n (Percentage)			*p* Value
		Low	Moderate	High	
Do you have vitamin D deficiency?	Yes	102 (74.45)	115 (53.48)	70 (54.68)	0.048
	No, or not sure	35 (25.54)	100 (46.51)	58 (45.31)	
Do you take vitamin D supplement or multivitamin that includes vitamin D?	Yes	98 (71.53)	45 (20.93)	39 (30.46)	0.017
	No	39 (28.46)	170 (79.06)	89 (69.53)	
Have you used sunscreen/sun protective cream in the last 12 months?	Yes	75 (54.74)	52 (24.18)	49 (38.28)	0.082
	No	62 (45.25)	163 (75.81)	79 (61.71)	
Did you ever try a tanning bed/tanning booth?	Yes	65 (47.44)	58 (26.97)	39 (30.46)	0.076
	No	72 (52.55)	157 (73.02)	89 (69.53)	

**Table 4 healthcare-11-02097-t004:** Prevalence of depression, anxiety, and stress among participants with vitamin D deficiency and control groups.

Categories	Vitamin D Deficient (N = 285)	Control (N = 193)	Total (N = 480)	*p* Value *
Depression (n)	172	92	264	0.042
Prevalence (%)	60.35%	47.66%	55%	
Anxiety (n)	189	103	292	0.031
Prevalence (%)	66.31%	53.36%	60.83%	
Stress (n)	214	112	326	0.001
Prevalence (%)	75.08%	58.03%	67.91%	

* Pearson Chi-square test (2-sided).

**Table 5 healthcare-11-02097-t005:** Level of severity of depression, anxiety, and stress among participants with vitamin D deficiency and control groups.

Categories	Vitamin D Deficient	Control	Total	*p* Value *
Depression, n (%)	172 (65.15)	92 (34.84)	264 (100)	0.032
Mild	51 (29.65)	42 (45.65)	93 (35.22)	
Moderate	87 (50.58)	43 (46.73)	130 (49.24)	
Severe	34 (19.76)	7 (7.60)	41 (15.53)	
Anxiety, n (%)	189 (64.72)	103 (35.27)	292 (100)	0.021
Mild	82 (43.38)	42 (40.77)	124 (42.46)	
Moderate	81 (42.85)	55 (53.39)	136 (46.57)	
Severe	26 (13.75)	6 (5.82)	32 (10.95)	
Stress, n (%)	214 (65.64)	112 (34.35)	326 (100)	0.011
Mild	64 (29.90)	54 (48.21)	118 (36.19)	
Moderate	116 (54.20)	44 (39.28)	160 (49.07)	
Severe	34 (15.88)	14 (12.5)	48 (14.72)	

* Pearson Chi-square test (2-sided).

**Table 6 healthcare-11-02097-t006:** Logistic regression analysis of factors associated with depression in study participants.

Categories	Odds Ratio	Confidence Interval (95%)		*p* Value
		Lower	Upper	
Vitamin D deficient	4.96	2.22	6.78	0.001
Female	3.45	1.99	4.14	0.011
Experience muscle pain	3.21	2.490	5.01	0.021
Frequent tiredness	2.98	1.311	4.61	0.011
Excessive academic demand	2.87	1.76	3.98	0.001
Family history of psychological burden	2.76	1.98	4.02	0.025
Obesity	2.67	1.87	4.32	0.034
Use of isotretinoin	2.34	1.88	4.56	0.022
Age between 20 and 22 years	1.67	1.23	2.87	0.039
Study level (1–4)	1.32	1.01	2.43	0.028
Use of statins	0.342	0.121	0.98	0.013

**Table 7 healthcare-11-02097-t007:** Logistic regression analysis of factors associated with anxiety in study participants.

Categories	Odds Ratio	Confidence Interval (95%)		*p* Value
		Lower	Upper	
Family history of psychological burden	4.55	2.25	8.65	0.001
Frequent tiredness	4.21	2.56	7.98	0.001
Use of isotretinoin	4.01	2.87	7.76	0.001
Vitamin D deficient	3.87	2.55	6.59	0.001
Excessive academic demand	3.56	2.11	7.65	0.001
Experience muscle pain	3.22	2.05	6.85	0.001
Female	3.21	2.15	6.87	0.001
Obesity	3.11	2.11	5.96	0.001
Age between 20 and 22 years	2.13	1.85	4.21	0.032
Study level (1–4)	1.32	1.04	2.76	0.022
Use of statins	0.54	0.23	0.98	0.019
Use of benzodiazepine	0.44	0.18	0.76	0.019
Use of beta-blockers	0.32	0.10	0.65	0.010

**Table 8 healthcare-11-02097-t008:** Logistic regression analysis of factors associated with stress in study participants.

Categories	Odds Ratio	Confidence Interval (95%)		*p* Value
		Lower	Upper	
Family history of psychological burden	5.67	2.071	9.32	0.001
Use of isotretinoin	4.87	2.98	8.64	0.001
Vitamin D deficient	4.77	3.21	9.33	0.001
Obesity	4.21	2.52	8.48	0.001
Frequent tiredness	3.44	2.11	9.21	0.001
Excessive academic demand	3.21	1.98	8.61	0.001
Experience muscle pain	2.45	1.21	6.54	0.001
Female	2.33	1.76	5.43	0.001
Study level (1–4)	2.31	1.73	6.21	0.001
Age between 20 and 22 years	1.32	0.98	3.22	0.032
Use of statins	0.89	0.32	1.34	0.021
Use of benzodiazepine	0.87	0.43	1.81	0.011
Use of beta-blockers	0.56	0.23	1.01	0.034

## Data Availability

Data is contained within the article.

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
