# Peer review of "Impact of Vitamin D Deficiency on Mental Health in University Students: A Cross-Sectional Study"

_healthcare, 2023, doi:10.3390/healthcare11142097_

Round 1
Reviewer 1 Report
The manuscript entitled “IMPACT OF VITAMIN D DEFICIENCY ON MENTAL HEALTHIN UNIVERSITY STUDENTS: A CROSS-SECTIONAL STUDY.” reported by Almuqbil et al., have studied the effect of Vit D deficiency in university students. There are few points need to be addressed by authors.
1. Authors claimed that the students pursuing education in health sciences disciplines are vulnerable to psychological burdens such as depression, anxiety, and stress. Did authors also perform the same study on non-science background students?
2. What are the reasons for depression, anxiety, and stress for health sciences?
3. What is the rational for choosing study survey time during March to May? Is this semester/ exam time for students?
4. Is Vit D deficiency leads depression, anxiety, and stress or vice-versa?
5. Its very hard to understand P-value applied by authors. For example, in table 1 section college, sub section- College of Medicine, College of Pharmacy, and College of Applied Science. For all three P-values is 0.078. P value is the comparison between two groups. Explain.
6. In discussion section authors used different font size. Correct it.
Author Response
Reviewer 1
Comments and Suggestions for Authors
The manuscript entitled “IMPACT OF VITAMIN D DEFICIENCY ON MENTAL HEALTHIN UNIVERSITY STUDENTS: A CROSS-SECTIONAL STUDY.” reported by Almuqbil et al., have studied the effect of Vit D deficiency in university students. There are few points need to be addressed by authors.
- Authors claimed that the students pursuing education in health sciences disciplines are vulnerable to psychological burdens such as depression, anxiety, and stress. Did authors also perform the same study on non-science background students?
Response: We appreciate your comment. This statement is not based on our study outcome, however, it’s on the basis on earlier studies. We agree with the comment of respected reviewer, since we have not done any comparison, we have corrected this statement in the Abstract.
- What are the reasons for depression, anxiety, and stress for health sciences?
Response: Now we have corrected this statement, our study focus is more towards the psychological burden in the university students irrespective of whether they are health sciences or non-health sciences university students.
- What is the rational for choosing study survey time during March to May? Is this semester/ exam time for students?
Response: Respected reviewer, this time is a slot during the second semester of the academic year where most of the students were available on campus as this corresponds to the time when transformation was taking place from online to on-campus education system due to COVID-19.
- Is Vit D deficiency leads depression, anxiety, and stress or vice-versa?
Response: Since we measured vitamin D and then evaluated the level of psychological burden and also there is high number of vitamin D deficient individuals found with psychological burden, it’s possible that the vitamin D deficiency has caused psychological burden.
- Its very hard to understand P-value applied by authors. For example, in table 1 section college, sub section-College of Medicine, College of Pharmacy, and College of Applied Science. For all three P-values is 0.078. P value is the comparison between two groups. Explain.
Response: Thank you so much respected reviewer. The comparison was done by non-parametric Chi-square test where comparison were done among more than three groups. Please find below reference:
Hazra A, Gogtay N. Biostatistics Series Module 4: Comparing Groups - Categorical Variables. Indian J Dermatol. 2016 Jul-Aug;61(4):385-92. doi: 10.4103/0019-5154.185700. PMID: 27512183; PMCID: PMC4966396.
- In discussion section authors used different font size. Correct it.
Response: Corrected now, thanks for your observation.

Reviewer 2 Report
The data that the authors have gathered are well presented, statistical analysis is ok and the conclusions are in accordance with the results. However I have a more general consideration concerning the classification of the participants to vitamin D deficient and control group since:
The classification is not based on actual measurement of vitamin levels but the participants classify themselves as deficient or not deficient. How do they "prove" their deficiency?
The time of the year (summer, winter etc) is an extra factor of question that could differentiate the answers of the participants.
Thus I believe that the authors should provide more information about the answers to the specific questions by the participants (were they informed about the normal and deficiency levels, how recent is the measurement etc.).
L49 "Vitamin D can be obtained in large amounts by sun exposure" . Vit D s not obtained by sun exposure. Provitamin D (7-DHC) is converted to previtamin D in the skin by exposure to UVB radiation and then previtamin D3 is isomerized by body heat to form vitamin D3 etc (Holick, M.F. Vitamin D: the underappreciated D-lightful hormone that is important for skeletal and cellular health. Multihormonal systems disorders. Curr. Opin. Endocrinol. Diabetes. 2002, 9, 87-98) Please rephrase.
Table 4: correct "prevalence" (inside table)
Qualty of English is acceptable.
Author Response
Reviewer 2
Comments and Suggestions for Authors
The data that the authors have gathered are well presented, statistical analysis is ok and the conclusions are in accordance with the results. However I have a more general consideration concerning the classification of the participants to vitamin D deficient and control group since:
The classification is not based on actual measurement of vitamin levels but the participants classify themselves as deficient or not deficient. How do they "prove" their deficiency?
Response: Thank you very much respected reviewer for your comment.
We have taken into consideration the vitamin D deficiency based on the vitamin D level measurement in addition to the declaration of the students that they are vitamin D deficient.
The details on these aspects are already included in section 2.2.
The time of the year (summer, winter etc) is an extra factor of question that could differentiate the answers of the participants.
Response: We concur with the esteemed critic. We conducted our research from March to May of 2021, which is the start of the summer. The research area's temperature is generally always between 25 and 35 degrees Celsius, and the sun is not as intense as it was from June to August. But it will be fascinating to gather information at other times of the year, compare vitamin D levels, and see how they relate to psychological stress. This will undoubtedly be covered in our upcoming research.
This has been included in the limitation of the study.
Thus I believe that the authors should provide more information about the answers to the specific questions by the participants (were they informed about the normal and deficiency levels, how recent is the measurement etc.).
Response: As stated in section 2.2, samples were collected from study participated to check the level of vitamin D based on which they were termed as vitamin D deficient or control. Therefore the measurement of vitamin D level was exactly at the time of doing the study.
L49 "Vitamin D can be obtained in large amounts by sun exposure" . Vit D s not obtained by sun exposure. Provitamin D (7-DHC) is converted to previtamin D in the skin by exposure to UVB radiation and then previtamin D3 is isomerized by body heat to form vitamin D3 etc (Holick, M.F. Vitamin D: the underappreciated D-lightful hormone that is important for skeletal and cellular health. Multihormonal systems disorders. Curr. Opin. Endocrinol. Diabetes. 2002, 9, 87-98) Please rephrase.
Response: It is rephrased now. Thanks for your suggestion.
Table 4: correct "prevalence" (inside table)
Response: Corrected now
Comments on the Quality of English Language
Qualty of English is acceptable.
Response: Thanks

Round 2
Reviewer 2 Report
The corrections and clarifications that the authors provide to my comments are sufficient.
Check again manuscript during proofreading for grammatical errors.